# Development of Novel Inhibitors Targeting the D-Box of the DNA Binding Domain of Androgen Receptor

**DOI:** 10.3390/ijms22052493

**Published:** 2021-03-02

**Authors:** Mariia Radaeva, Fuqiang Ban, Fan Zhang, Eric LeBlanc, Nada Lallous, Paul S. Rennie, Martin E. Gleave, Artem Cherkasov

**Affiliations:** Vancouver Prostate Centre, University of British Columbia, 2660 Oak Street, Vancouver, BC V6H 3Z6, Canada; mradaeva@prostatecentre.com (M.R.); fban@prostatecentre.com (F.B.); fzhang@prostatecentre.com (F.Z.); eleblanc@prostatecentre.com (E.L.); nlallous@prostatecentre.com (N.L.); prennie@prostatecentre.com (P.S.R.); m.gleave@ubc.ca (M.E.G.)

**Keywords:** prostate cancer, computer-aided drug discovery, small-molecule inhibitors, androgen receptor, dimerization

## Abstract

The inhibition of the androgen receptor (AR) is an established strategy in prostate cancer (PCa) treatment until drug resistance develops either through mutations in the ligand-binding domain (LBD) portion of the receptor or its deletion. We previously identified a druggable pocket on the DNA binding domain (DBD) dimerization surface of the AR and reported several potent inhibitors that effectively disrupted DBD-DBD interactions and consequently demonstrated certain antineoplastic activity. Here we describe further development of small molecule inhibitors of AR DBD dimerization and provide their broad biological characterization. The developed compounds demonstrate improved activity in the mammalian two-hybrid assay, enhanced inhibition of AR-V7 transcriptional activity, and improved microsomal stability. These findings position us for the development of AR inhibitors with entirely novel mechanisms of action that would bypass most forms of PCa treatment resistance, including the truncation of the LBD of the AR.

## 1. Introduction

Prostate cancer (PCa) is the most prevalent malignancy in men and the second most common cause of cancer-related death in Canada and the United States (US) among males [1]. The androgen receptor (AR), a ligand-inducible transcription factor, plays a critical role in the development of PCa, where it controls the transcription of onco-driving genes [2,3,4,5,6]. The inhibition of androgen signaling remains one of the major strategies for PCa treatment; the early stages of PCa respond well to androgen deprivation therapy, surgery, and radiation. Unfortunately, most patients eventually develop a more aggressive and lethal androgen-independent form of the disease called castration-resistant PCa (CRPC) that arises through various mechanisms, including gain-of-function mutations in the AR gene and increased AR expression [7,8,9].

The development of more recent antiandrogen drugs, such as enzalutamide and abiraterone, that competitively bind the ligand-binding domain (LBD) of AR helped decrease PCa mortality and morbidity significantly [10,11,12,13]. However, resistance to these drugs could also eventually develop by diverse means, including LBD mutations and the occurrence of splice variants in the AR [9,14,15,16]. In particular, the CRPC-prevalent AR-V7 splice variant completely lacks the LBD and becomes resistant to all the LBD-binding drugs, which comprise the whole spectrum of clinically used antiandrogens [17]. Hence, the development of LBD-independent drugs becomes an attractive strategy to combat advanced forms of PCa [18].

In previous studies [18,19,20], we have demonstrated that the DNA-binding domain (DBD) located between the N-terminal domain (NTD) and the LBD is an attractive drug target with significant therapeutic potential [11] (Figure 1A). The DBD folds consist of two α-helixes: P-box ‘recognition helix’ that directly binds transcription factor motifs of the downstream-regulated genes and D-box that is responsible for AR dimerization [6,21]. We previously reported small molecule inhibitors that bind the P-box and, hence, directly block the AR-DNA interactions (Figure 1B), which proves the feasibility of targeting the AR DBD [18,19,20]. Another strategy to target the AR DBD would be to disrupt its homo-dimerization, which is essential for the activation of all forms of AR, including full-length and splice variants [22,23,24,25]. Indeed, while monomers of full-length AR could still interact with androgen response elements to a minimal extent, monomers of AR-V7 seem to be completely non-functional [22].

In a previous study [26], we identified a druggable binding site (Figure 1C) on the surface of the D-box and developed small molecules that disrupt the AR functional dimerization. Our lead P-box inhibitor VPC-17005 suppressed androgen signaling, demonstrated AR specificity through mutagenesis studies, and inhibited the growth of AR-positive cancer cell lines. However, VPC-17005 demonstrated rather poor metabolic stability making it a non-suitable candidate for in vivo studies and future optimization. Thus, we decided to expand the repertoire of AR DBD-dimerization inhibitors and evaluate more chemotypes by using a combination of docking, pharmacophore modeling, and a diverse set of biochemical assays.

Herein, we report the newly discovered series of chemically diverse small molecules that were characterized in numerous AR-relevant assays and shown to abrogate DBD dimerization effectively. In particular, two lead compounds, VPC-17160 and VPC-17281, demonstrated stronger inhibition of AR-V7 transcriptional activity and exhibited superior activity in mammalian two-hybrid assays compared to our previously identified lead (VPC-17005). Furthermore, compound VPC-17281 appeared to have significantly improved microsomal stability, which is critical for future hit-to-lead optimization. Overall, these findings provide insights on the inhibition of DBD dimerization and provide a foundation for further development of AR-targeting drugs that bypass most of the known mechanisms of PCa drug resistance.

## 2. Results

### 2.1. In Silico Identification of VPC-17160 and VPC-17281

Our earlier lead VPC-17005 demonstrated good AR inhibition but sub-optimal pharmacokinetic properties with low microsomal stability (t_1/2_ = 14 min). We utilized the structure of rat AR DBD dimer bound to the DR3 oligonucleotide (PDB (Protein Data Bank) code: 1R4I) to dock drug-like compounds from the ZINC15 database into the previously identified binding site [21,27]. We then developed a pharmacophore model based on the binding pose of VPC-17005 (Appendix A). The pharmacophore model consists of one hydrogen-bond acceptor and one aromatic ring feature, as these interactions were noted to be crucial in our previous study. We then selected 240 compounds based on this pharmacophore model that were purchased and tested for AR inhibition activity.

### 2.2. VPC-17160 and VPC-17281 Reduces AR-FL and AR-V7 Transcriptional Activities

Compounds were first screened for their inhibition of AR-V7 transcriptional activity. To do so, we created a PC3 cell-based population that stably encodes (i) a doxycycline-inducible AR-V7 cDNA sequence (pLIX402 tet-inducible promoter, puromycin selection), and (ii) the ARR3tk-nanoLuciferase reporter construct. Hence, nanoluciferase production was directly correlated with AR-V7 activity. This allowed the screening of the compounds without using the error-prone method of transiently transfecting V7 and reporter constructs. The inhibition efficacy of compounds was measured using doxycycline treatment along with an increasing dose of compounds. From this screen, a panel of compounds was identified with some anti-dimerization activity (Appendix A). Two hit compounds, VPC-17160 and VPC-17281, showed complete AR-V7 inhibition at 10 μM (Figure 2D). To rule out direct effects of the compounds on luciferase expression or PC3 cell viability, we used a PC3-dox inducible luciferase cell line in parallel. Using a concentration-dependent titration, we next established that their IC_50_ values for the inhibition of the AR-V7 transcription were 6 μM for both compounds (Figure 2A). This data indicates a 2-fold improvement in the potency of AR-V7 inhibition in comparison with the parental compound VPC-17005 (IC_50_ = 10 µM).

In addition, these compounds were also tested for their ability to inhibit full-length AR transcriptional activity in LNCaP cells using an enhanced green fluorescent protein (eGFP) AR transcriptional assay, where the expression of eGFP is under the direct control of an androgen-responsive probasin-derived promoter [28]. Compounds VPC-17160 and VPC-17281 showed significant inhibition of AR-FL expression at 12 μM concentration with estimated IC_50_ of 2 and 5 μM, respectively (Figure 2B and Appendix A).

To further validate compounds as AR inhibitors, we tested their activity by quantifying the effect on the production of endogenous prostate-specific antigen (PSA) in PCa cell lines. PSA is an AR-regulated serine protease and is widely used as a biomarker for PCa [29]. As expected, VPC-17160 and VPC-17281 induced a significant decrease in secreted PSA levels in LNCaP cells with corresponding IC_50_ values established at 2 μM and 6 μM, respectively (Figure 2C).

### 2.3. Evidence for an Anti-Dimer Mechanism

To validate that this activity occurred via anti-homodimer inhibition rather than androgen displacement at the LBD, we measured the ability of VPC-17160 and VPC-17281 to displace a fluorescent-tagged androgenic ligand (FluormoneAL Green) from the AR-LBD. Even at high concentrations (50 μM), no significant hormone displacement was observed for compound VPC-17281 (Table 1). We observed around 25% of displacement for compound VPC-17160 at 10 μM, a concentration higher than the IC_50_ of the compound in transcription assays. In contrast, parental compound VPC-17005 was displacing 75% of the androgen at 10 μM, suggesting that this compound was having some of its efficacy through the ligand-binding site. This would explain why VPC-17160 and VPC-17281 are more potent inhibitors than VPC-17005 in the AR-V7 transcription assay.

Next, the dimerization of full-length AR was assessed by a mammalian two-hybrid (M2H, [30]) assay, where target proteins were fused with Gal4-DBD or VP16 domains. Co-expression of Gal4-DBD-AR (pB-AR) and VP16-AR (pA-AR) resulted in luciferase activity from the Gal4 responsive PG5-NLuc reporter, signifying dimerization (Figure 3). High concentrations of VPC-17005 (>20 μM) were required for significant reduction in AR and AR-V7 homo-dimerization. In contrast, VPC-17160 and VPC-17281 induced full reduction of dimerization at 20 μM concentration, suggesting a better efficacy of these compounds to inhibit dimerization.

### 2.4. Effect on the Growth of Prostate Cancer Cell Lines

VPC-17160 and VPC-17281 were further evaluated for their potential to inhibit the growth of prostate cancer cell lines LNCaP (AR-FL dependent) and 22rv1 (AR-V7 dependent). As anticipated, all compounds effectively inhibited the growth of the AR-dependent LNCaP cell line. Compound VPC-17281 showed an IC_50_ of 25 μM in the LNCaP cell line. Interestingly, compound VPC-17281 caused a significant reduction in the growth of 22rv1 cells (IC_50_ = 10 μM; see Figure 4), which suggests that this compound truly targeted the truncated variant of the AR.

We then tested the metabolic stability of these molecules using human microsomes. Additionally, the measured half-life in microsomes increased by approximately six-fold for compound VPC-17281 (T_1/2_ > 90 min) compared to VPC-17005 (T_1/2_ = 14 min), whereas compound VPC-17160 showed poor stability with a half-life of 3 min. This could explain the lower potency of this compound in cell growth assays, as it is metabolized quickly in the media.

## 3. Discussion

Prostate cancer growth is primarily driven by AR signaling, which could be effectively depleted with androgen-competing chemicals. All drugs currently on the market act through binding to the ligand-binding domain (LBD) of the AR. Unfortunately, a large number of patients develop resistance to androgen displacement treatment. Thus, the development of drugs that target a different site on the surface of AR (such as the dimerization site) remains an attractive strategy to combat drug resistance.

Previously we reported VPC-17005, a small molecule inhibitor that targets the dimerization interface on the AR DBD that demonstrated promising AR inhibition in androgen-sensitive LNCaP and enzalutamide-resistant MR49F cells in vitro [26]. However, further use of VPC-17005 was limited by poor metabolic stability and limited potency against the AR-V7. Thus we have utilized the structure of VPC-17005 as a template to develop further improved AR dimerization inhibitors that demonstrate enhanced target affinity and drug-like properties. From this screen combining docking and pharmacophore modeling with wet-lab experiments, we identified two leads—VPC-17281 and VPC-17160—that outperformed the parent compound. These new AR DBD P-box inhibitors are likely to be more specific as their cross-interaction with the LBD site was minimal (according to the DHT displacement assay). Furthermore, this observation was in agreement with the enhanced activity of VPC-17281 and VPC-17160 against the AR-V7 truncated form of the receptor. The docking models we developed indicate that the enhanced affinity towards the DBD dimerization site could result from a larger hydrophobic surface area of the newly developed compounds and the increased number of hydrophobic contacts with the protein (Figure 5). Other additional protein–ligand interactions include a hydrophobic contact formed with Leu595 by both VPC-17160 and VPC-17281 but not VPC-170005 (Figure 5). In addition, these molecules are generally bulkier than the parent compound, and, thus, they are less likely to be accommodated by the ligand-promiscuous LBD site. The higher affinity of VPC-17281 for the receptor could also be justified by the formation of an additional hydrogen bond between the hydroxyl group and Asn611 as predicted by the binding pose (Figure 5). Overall, the developed pharmacophore model successfully identified compounds that bind the dimerization site and could be used for further screenings.

Furthermore, both lead compounds VPC-17160 and VPC-17281 inhibited the growth of 22rv1 and LNCaP cancer cell lines, proving the antineoplastic effects of the dimer inhibition. Importantly, the profound activity of VPC-17281 on the AR-V7 dependent 22rv1 cell line provides additional assurance as to the effectiveness of our lead inhibitor and its applicability for targeting castration-resistant prostate cancer cells. However, it is important to mention that VPC-17281 also demonstrated some activity in the PC3 cell line, which suggests off-target effects and possible toxicity. In contrast, VPC-17160 showed no effect on the PC3 cell lines suggesting its selectivity to AR and could represent the best starting template for further drug optimization.

It should also be noted that microsomal stability represents one of the key pharmacokinetic properties of a candidate drug since a low microsomal half-life usually indicates poor bioavailability. VPC-17281 showed a greatly improved microsomal stability, while VPC-17160 was readily metabolized in microsomes. These results indicate that further optimization of the compounds is necessary to overcome the issues of toxicity and low microsomal stability.

## 4. Methods

### 4.1. In Silico Identification Experiments

The crystal structure of rat AR DBD was downloaded from PDB database (1R4I) and prepared with Protein Preparation Wizard from Schrödinger [31,32]. The preparation included the addition of hydrogen atoms (followed by energy minimization), assignment of bond orders, and filling of the missing side chains. Compounds were derived from the ZINC15 database and then prepared with MOE (Molecular Operating Environment) [27,33]. In particular, molecules were protonated, partial charges were added, and then they were subjected to energy minimization. The docking was performed with GlideSP from Schrödinger [34]. Pharmacophore model development and screening were also performed using MOE.

### 4.2. Compounds

All compounds were identified from the ZINC15 database and purchased from Enamine (https://enamine.net/, Accessed 1 March 2021).

### 4.3. Plasmids and Constructructs

Full-length human AR (hARWT) was encoded on pcDNA3.1 expression plasmid. Lentivirus plasmid pLIX402-ARV7 (ARV7 cDNA, doxycycline-inducible promoter, puromycin selection) was a gift from Dr. N. Lack. pLIX402-Nanoluc was constructed by excising the ARV7 cDNA from pLIX402-ARV7 (SalI and BamHI) and ligating Nanoluciferase (Nanoluc or NLuc) cDNA amplified from pNL1.1(CMV) plasmid. Lentivirus plasmid encoding the AR-responsive reporter was created by replacing the Ubiquitin C promoter in pLB-U [35] with the ARR3tk promoter (3 × probasin, 23) using PacI/NheI restriction sites on pLB-U plasmid obtained from Dr. C. Ong. Gateway cloning was employed to insert Nanoluc cDNA to yield pLB-ARR3tk-NLuc. Full-length AR cDNA were cloned using Polymerase Incomplete Primer Extension (PIPE [36]), into CheckMate™ pAct (pA, VP16) or pBIND (pB, Gal4-DBD) constructs (Promega, Madison, WI, USA). Similarly, PG5-NLuc was constructed by replacing firefly luciferase from PG5-Luc (Promega) with Nanoluc cDNA using PIPE.

### 4.4. PCa Cell Lines

LNCaP (ATCC, CRL-1740), 22rv1 (ATCC, CRL-2505), and PC3 (ATCC, CRL-1435) cells were obtained in 2013 and authenticated by IDEXX Laboratories in 2014 every 6 months and tested for mycoplasma contamination every two weeks.

### 4.5. Androgen Displacement Assay

Androgen displacement was assessed with the Polar Screen Androgen Receptor Competitor Green Assay Kit as per the instructions of the manufacturer.

### 4.6. Reporter Assays and Cell Viability Experiments

LNCaP cells incorporating an AR2PB-eGFP reporter construct (2 × probasin) are previously described [28]. eGFP and secreted PSA assays were performed as described [20,28]. The mammalian two-hybrid assay was performed in PC3 cells by co-transfection of 15 ng pA-AR or pB-AR plasmids, with 12.5 ng PG5-NLuc reporter. PC3_iV7_3TKNluc or PC3_iNluc cells were seeded in DMEM +5% CSS at 5000 cells/well for 24 h, followed by adding doxycycline/compounds (18 h) and luminescence measurement. LNCaP and 22rv1 cells were starved in RPMI + CSS (48 h) before cell viability tests using the same conditions as above, followed by 30 μL/well PrestoBlue™ Reagent (1 h) and fluorescence measurement (TECAN F200).

### 4.7. Statistical Analysis

All statistical analysis and dose–response curves were generated using GraphPad Prism 6.07 software (GraphPad Software, La Jolla, CA, USA), www.graphpad.com (accessed 1 March 2021).

## 5. Conclusions

In the current study, we discovered novel inhibitors of AR DBD dimerization through the synergetic application of virtual screening methods and experimental validation. The study was built upon a previously reported AR DBD inhibitor VPC-17005 that suffered from sub-optimal biostability and pharmacokinetics. The newly developed lead compounds, VPC-17281 and VPC-17160, belong to different chemotypes and exhibit significantly improved microsomal stability and anti-AR activity. This study broadens the repertoire of compounds targeting AR DBD dimerization and could provide useful insights for the further development of novel antiandrogens with unique and novel mechanisms of action. These drugs would be of immense value for CRPC patients that do not respond to conventional AR inhibitors.

## Figures and Tables

**Figure 1 ijms-22-02493-f001:**
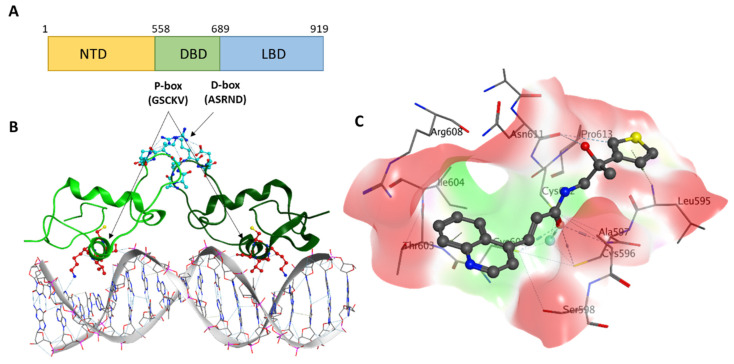
(**A**) Androgen receptor (AR) domain structure with P-box and D-box residues shown. (**B**) Crystal structure of the rat AR DNA binding domain (DBD) dimer (monomers are shown in different shades of green) bound to DNA (PDB (Protein Data Bank) code: 1R4I). The P-box residues are shown in blue, while D-box residues are in red. (**C**) Stick diagram model of VPC-17281.

**Figure 2 ijms-22-02493-f002:**
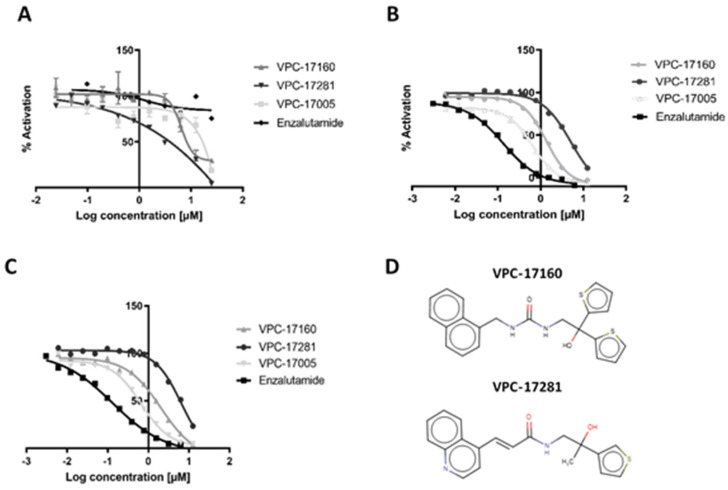
In vitro inhibition of AR-V7 and AR-FL transcription. (**A**) Dose-response curve illustrating the inhibition of AR-V7 in doxycycline-inducible AR-V7 construct with ARR3tk-nanoLuciferase reporter, (**B**) Inhibition of AR transcriptional activity in LNCaP-eGFP cells, (**C**) The inhibition of AR-mediated prostate-specific antigen (PSA) expression in LNCaP cells. Data points represent a pool of triplicates for each concentration. (**D**) Chemical structure of VPC-17160 and VPC-17281.

**Figure 3 ijms-22-02493-f003:**
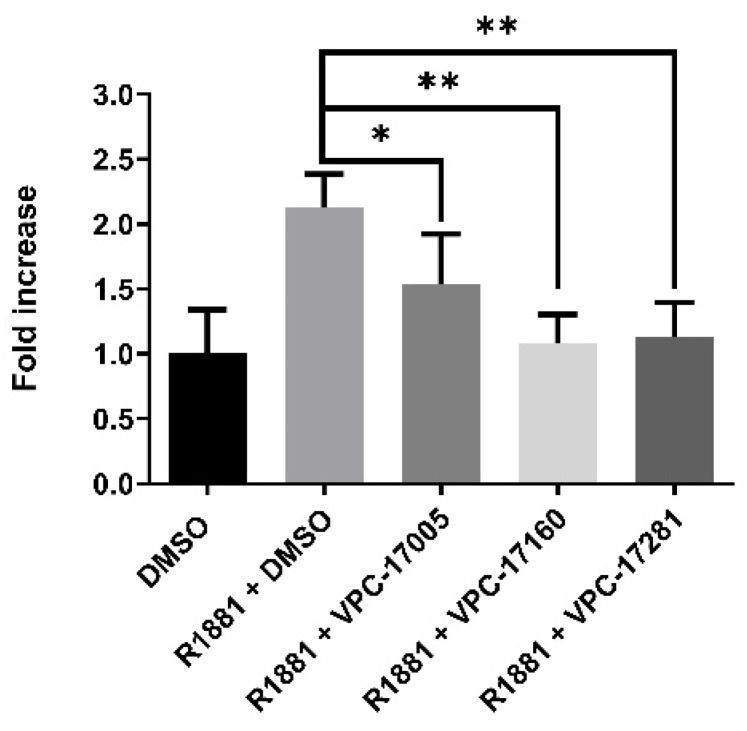
Anti-dimerization effects of VPC-17160 and VPC-17281 at 20 μM concentraction. PC3 cells were co-transfected with plasmids encoding pA-AR and pB-AR and pG5-NLuc-Gal4 specific reporter. Cells were treated with 0.1% DMSO control, VPC-17160, VPC-17281, or VPC-17005 followed by Nanoluciferase measurement. *p* < 0.05 (*), *p* < 0.01 (**) were considered statistically significant (two-tailed *t*-test). Two-tailed *t*-test for VPC-17005 vs. 17160 *p* < 0.05; VPC-17005 vs. VPC-17281 *p* < 0.05.

**Figure 4 ijms-22-02493-f004:**
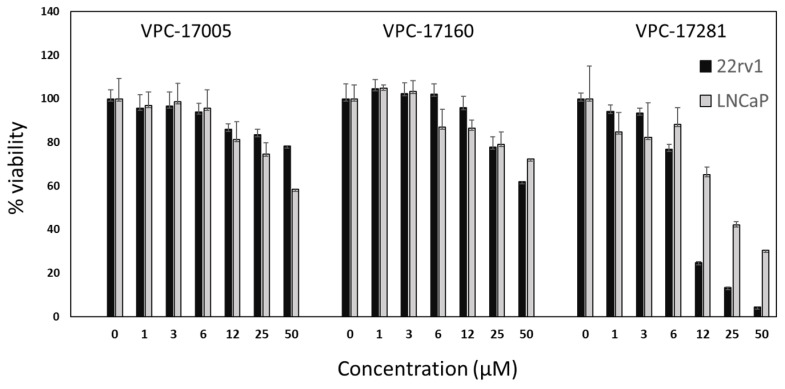
The effect of VPC-17160 and VPC-17281 on cell proliferation in LNCaP and 22rv1 cells. Percent cell viability is plotted in a dose-dependent manner. Data points represent a pool of triplicates for each concentration. All data are presented as mean ± SEM.

**Figure 5 ijms-22-02493-f005:**
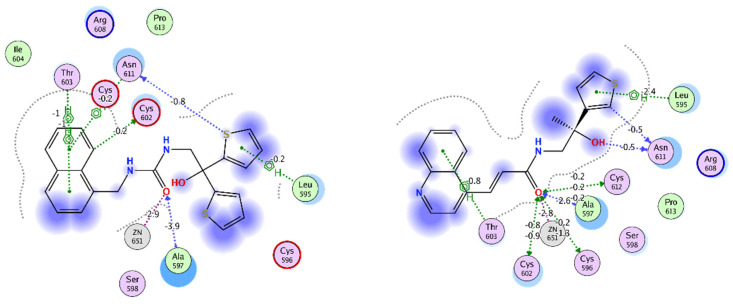
Ligand–protein interactions of VPC-17160 (**left**) and VPC-17281 (**right**). Residues in green—hydrophobic, in purple—polar. Numbers next to bonds indicate estimated bond energy in kcal/mol.

**Table 1 ijms-22-02493-t001:** Androgen displacement at the ligand-binding site and microsomal stability.

Compound	% Displacement		Half-Life (min)
	(1 μM)	(10 μM)	
Enzalutamide	63	85	>1000
VPC-17005	26	56	14
VPC-17160	10	29	3
VPC-17281	0	0	90

## Data Availability

Data is contained within the article and Appendix A.

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
