# Peer review of "Development of Novel Inhibitors Targeting the D-Box of the DNA Binding Domain of Androgen Receptor"

_ijms, 2021, doi:10.3390/ijms22052493_

Round 1

Reviewer 1 Report

The authors revealed that novel inhibitors (VPC-17160 and VPC-17281) targeting the D-BOX of the DNA binding domain of androgen receptors with appropriate experiments. Several results using different methods make this work interest. Although the studies are interesting, the authors should address the following minor concerns.

  1. Authors kindly include the IC50 values of VPC-17160 and VPC-17281 compound treated Lncap cells in the result section.(Figure 4)
  2. Authors kindly include the statistical analysis for Figure 3 and 4
  3. Please include the VPC-17160 and VPC-17281 concentrations in Anti-dimerization assay in the result section.(Figure 3)
  4. Authors kindly remove the LnCap cell viability figure for VPC-17005 compound (already published)
  5. The authors did check the effect cell viability for VPC-17160 and VPC-1728 compound in AR null cell line?
  6. Did authors check AR nuclear protein downregulation(western or immunofluorescence assay) in VPC-17160 and VPC-17281 treated Lncap or 22RV1 cells ?

Author Response

  1. Authors kindly include the IC50 values of VPC-17160 and VPC-17281 compound treated Lncap cells in the result section.(Figure 4)

We thank the reviewer for the suggestion. We added the IC50 value for VPC-17281 in the LNCaP cell line. Unfortunately, an IC50 value for VPC-17160 could not be determined precisely because the curve appear to be incomplete.

  1. Authors kindly include the statistical analysis for Figure 3 and 4

We included statistical significance on the figure and added statistical analysis to the figure legend.

  1. Please include the VPC-17160 and VPC-17281 concentrations in Anti-dimerization assay in the result section.(Figure 3)

We added the concentration used for both compounds – 20 μM.

  1. Authors kindly remove the LnCap cell viability figure for VPC-17005 compound (already published)

We thank the reviewer for the suggestion, however, we believe that this figure is essential as it is used for the comparison (VPC-17005 is the parent compound).

  1. The authors did check the effect cell viability for VPC-17160 and VPC-1728 compound in AR null cell line?

As we mentioned in the Discussion, toxicity was observed with compound VPC-17281 in PC3 cells after 3 days of growth. However, a nanoluciferase counterscreen was run alongside with the PC3-V7 assay to rule out that lowering nanoluciferase signal after 24 hours was due to cell death.

  1. Did authors check AR nuclear protein downregulation (western or immunofluorescence assay) in VPC-17160 and VPC-17281 treated Lncap or 22RV1 cells ?

We thank the reviewer for the comment, but we decided to proceed with such analysis with better and more stable derivatives of these compounds.

Reviewer 2 Report

The manuscript by Radaeva et al. focuses on development of small-molecule inhibitors of AR DBD dimerization through virtual screening and experimental validation.  The work is an extension of authors’ previous reports where they identified an AR DBD inhibitor VPC-17005 that possessed sub-optimal microsomal stability and pharmacokinetics. The newly identified lead compounds VPC-17281 and VPC-17160 belong to different chemotypes and exhibit improved activity in the mammalian two-hybrid assay, enhanced inhibition of AR-V7 transcriptional activity and improved microsomal stability. The manuscript, in general, is well presented and may be published in IJMS after addressing the following minor issues:

How were the IC50 values calculated?

Page 1, lines 38-40 – Change ‘LDB’ to ‘LBD’

Enzalutamide is referred by its alternative name MDV3100 in Fig. 2B-D. Change it to enzalutamide to be consistent or define the terminology in the text at first use.

Page 2, line 42- rephrase this sentence

Author Response

  1. How were the IC50 values calculated?

We thank the reviewer for the suggestion. Dose–response curves were generated using GraphPad Prism 6.07 software (GraphPad Software, La Jolla, CA, USA). We added this information to the methods section.

  1. Page 1, lines 38-40 – Change ‘LDB’ to ‘LBD’

We appreciate this comment; the typo was fixed.

  1. Enzalutamide is referred by its alternative name MDV3100 in Fig. 2B-D. Change it to enzalutamide to be consistent or define the terminology in the text at first use.

We thank the reviewer for mentioning this. We changed the figure labels.

  1. Page 2, line 42- rephrase this sentence

We rephrased this sentence